# Prevalence of *Salmonella* in Free-Range Pigs: Risk Factors and Intestinal Microbiota Composition

**DOI:** 10.3390/foods10061410

**Published:** 2021-06-18

**Authors:** Victoria Garrido, Lourdes Migura-García, Inés Gaitán, Ainhoa Arrieta-Gisasola, Ilargi Martínez-Ballesteros, Lorenzo Fraile, María Jesús Grilló

**Affiliations:** 1Animal Health Group, Instituto de Agrobiotecnología (CSIC-Gobierno de Navarra), 31192 Mutilva, Navarra, Spain; victoria.garrido@csic.es (V.G.); inesgaitanmarqueta@gmail.com (I.G.); 2IRTA, Centre de Recerca en Sanitat Animal (CReSA, IRTA-UAB), OIE Collaborating Centre for the Research and Control of Emerging and Re-Emerging Swine Diseases in Europe, Campus Universitat Autònoma de Barcelona, 08193 Bellaterra, Barcelona, Spain; lourdes.migura@irta.cat; 3MikroIker Research Group, Immunology, Microbiology and Parasitology Department, Faculty of Pharmacy, University of the Basque Country (UPV/EHU), Paseo de la Universidad 7, 01006 Vitoria-Gasteiz, Spain; ainhoa.arrieta@ehu.eus (A.A.-G.); ilargi.martinez@ehu.eus (I.M.-B.); 4Departamento de Ciencia Animal, Universidad de Lleida, 25198 Lleida, Spain; lorenzo.fraile@udl.cat

**Keywords:** *Salmonella*, microbiota, free-range pigs, risk factors, antimicrobial resistance

## Abstract

Extensive pig systems are gaining importance as quality production systems and as the standard for sustainable rural development and animal welfare. However, the effects of natural foods on *Salmonella* epidemiology remain unknown. Herein, we assessed the presence of *Salmonella* and the composition of the gut microbiota in pigs from both *Salmonella*-free and high *Salmonella* prevalence farms. In addition, risk factors associated with the presence of *Salmonella* were investigated. The pathogen was found in 32.2% of animals and 83.3% of farms, showing large differences in prevalence between farms. Most isolates were serovars Typhimurium monophasic (79.3%) and Bovismorbificans (10.3%), and exhibited a multi-drug resistance profile (58.6%). Risk factor analysis identified feed composition, type/variety of vegetation available, and silos’ cleaning/disinfection as the main factors associated with *Salmonella* prevalence. Clear differences in the intestinal microbiota were found between *Salmonella*-positive and *Salmonella*-negative populations, showing the former with increasing *Proteobacteria* and decreasing *Bacteroides* populations. Butyrate and propionate producers including *Clostridium*, *Turicibacter*, Bacteroidaceae_uc, and *Lactobacillus* were more abundant in the *Salmonella*-negative group, whereas acetate producers like *Sporobacter*, *Escherichia* or *Enterobacter* were more abundant in the *Salmonella*-positive group. Overall, our results suggest that the presence of *Salmonella* in free-range pigs is directly related to the natural vegetation accessible, determining the composition of the intestinal microbiota.

## 1. Introduction

*Salmonella enterica* is one of the most common causes of food-borne zoonotic diseases worldwide. A total of 87,923 cases were reported in the European Union (EU) during 2019, making it the second most important zoonosis after campylobacteriosis [1]. Eggs and poultry products have been considered the most important source of human infections, responsible for 43.8% of cases [2], but the implementation of *Salmonella* control programs for fowl have resulted in decreasing trends in the occurrence of *Salmonella* in eggs in the EU. Consequently, a decrease in human salmonellosis has been reported from 2008 to 2019 [3]. Currently, pigs are the most important source of infection in humans, which has become a serious public health problem that requires special monitoring and surveillance. To preserve consumers’ health, the EU is discussing the establishment of restrictions in the international trade of pigs and pig products with countries that do not meet the objective of reducing the prevalence of *Salmonella*. These restrictions may have a major economic impact in our country because the pig sector is a key pillar of livestock resources in Spain, which is the fourth largest producer worldwide after the US, China and Germany [4]. Carrier animals are a serious food safety issue because they potentially shed the pathogen in feces, thereby contaminating other animals, slaughterhouses and meat products during processing. Additionally, the control of *Salmonella* in the food chain is impeded by the existence of over 2500 serovars, its broad host range and ubiquitous nature and its ability to sub-clinically colonize animals intended for human consumption.

In the recent years, consumers have become more aware of the way food is produced [5]. The increasing intensification of farming has been perceived negatively, while “animal-friendly” production such as free range is considered positive. Thus, extensive production has several features perceived by consumers as an improvement when compared with intensive management: (i) The animals mainly consume pasturage, reducing the need for industrial feed; (ii) It tends to raise autochthonous breeds that are well-adapted to the conditions of the land and extensive management; (iii) The production system is a sustainable model involving familiar farms; (iv) It provides the animals with the possibility of showing their natural behaviour; (v) It allows for low antibiotic usage; (vi) Due to natural food intake, the meat has a higher proportion of saturated/unsaturated fat, being healthier than intensive breeding; and (vii) The number of animals has to be kept in proportion with the available land due to the need for pastures, as well as slurry assimilation by the land and the excrements used as fertilizers. Extensive production promotes environmental sustainability and the development of rural areas.

The endogenous microbiota is known to provide important benefits to its host [6], and there is growing evidence suggesting that interactions between members of the gut microbiota of the host contribute to the health and well-being of these animals [7]. Recent studies have shown that *Salmonella enterica* is a pathogen capable of causing alterations to the composition of the intestinal microbiome, reducing the diversity and abundance of species regarded as beneficial such as lactic acid bacteria [8,9]. However, most of these studies are based in experiments where pigs have been challenged with the pathogen, and very few studies have focused on natural exposure of pigs to the pathogen; this is especially true for free range pigs. Thus, the main goal of this study is to widen the knowledge available on the microbiota of pigs raised in extensive systems in relation to the presence/absence of *Salmonella*, determine the prevalence of *Salmonella* spp. in this population, and decipher the risk factors associated with *Salmonella* spp. prevalence in pigs raised in outdoor systems.

## 2. Materials and Methods

### 2.1. Study Design

In order to study the presence of *Salmonella* in the intestinal contents (IC) of free-range fattening pigs (Duroc), 12 out of the 32 farms located in the Basque Country and north of Navarre were selected. They were visited between January and May 2015. The peculiarity of these farms was that animals were fed with cereals and food supplements *ad libitum* that were obtained directly from the field (pasture, acorns, chestnuts, or beechnuts).

All the animals of this study were transported to a slaughterhouse located less than 100 km (less than 2 h) from the farms, and they were slaughtered within 15–18 h of fasting. The intestinal contents (IC) of the cecum were collected from 15 animals/farm (*n* = 180 samples), randomly selected in the slaughterhouse line. The selection of cecum samples was based on the disruption to the microbiota in this anatomical location when challenged with *Salmonella* Typhimurium, as described by Borewicz et al. [9]. The whole intestinal package was removed from the selected carcasses at the evisceration point of the slaughter line, and 30 g of IC were collected aseptically in one-use sterile bottles, transported at 4 °C to the laboratory and immediately processed for *Salmonella* isolation (25 g). One gram per sample was frozen at −80 °C for microbiota analysis.

Ethics committee approval was not required because the work was performed with the entrails obtained in the conventional slaughtering line. Animal handling and slaughtering procedures were performed by the slaughterhouse personnel according to the current national legislation (Law 32/2007) for animal care regarding holding, transportation, testing and slaughtering.

### 2.2. Isolation of Salmonella

*Salmonella* isolation procedures were performed according to ISO 6579:2002/Amd 1:2007 (ISO) rules [10], as previously described [11,12,13]. All culture media and reagents used were purchased from Laboratorios Conda S.A., Spain. The IC samples (25 g) were homogenized in sterile filter bags (Stomacher^®^ 80, Seward Medical) with 225 mL of Buffer Peptone Water for pre-enrichment. After the different ISO steps, one colony of *Salmonella* was sent for serotyping at the National Reference Laboratory for Animal Salmonellosis, Central Veterinary Laboratory (LCV, Madrid, Spain), following the Kauffmann–White Scheme [14].

Next, all samples were analyzed using a specific *Salmonella* PCR-*invA* [15] with the DNA extracted by boiling (95 °C, 10 min) from a 10 µL loop of MRSV that was mixed with 500 µL ultra-pure sterile water and showed the halo typical of *Salmonella*. Thereafter, samples showing positive results by PCR-*invA* and negative by ISO were re-analyzed microbiologically for *Salmonella* isolation.

All isolates were serotyped at the reference LCV (see above) and stored in the IdAB-CSIC collection by freezing in sterile, 10% skim milk (BD Difco™ Skim Milk, Sparks, MD, USA) supplemented with 3% lactose (Merck, Billerica, MA, USA).

The agreement between microbiological ISO and PCR-*invA* diagnosis was calculated using the kappa (κ) index and was interpreted according to the criteria of Landis and Koch [16].

### 2.3. Antimicrobial Susceptibility Testing

Confirmed *Salmonella* isolates were tested by the Kirby–Bauer disk diffusion method, using the antimicrobials (BD Diagnostics) and concentrations recommended by the current EU legislation for the harmonized monitoring of the antimicrobial resistance of *Salmonella* [17], namely, ampicillin (A), chloramphenicol (C), streptomycin and gentamicin (S), sulfisoxazole and trimethoprim–sulfamethoxazole (Su), tetracycline (T), nalidixic acid (Nx), enrofloxacin and cefotaxime (C3G). Antimicrobial susceptibility was determined by measuring the inhibition halo generated after incubation (37 °C, 24 h). Strains *Escherichia coli* ATCC 25,922, *Salmonella* Typhimurium DT104 and ATCC 14028 were used as controls. Isolates were classified as resistant or susceptible according to Clinical and Laboratory Standard Institute (CLSI) recommendations [18]; those exhibiting resistance to drugs of three or more antimicrobial families were considered multi-drug resistant (MDR).

### 2.4. Epidemiological Study and Risk Factors

Risk factors associated with *Salmonella* presence in pigs’ IC were studied by performing an epidemiological survey, with 73 variables divided into five sections. The variables analyzed were related to the geographical location of farms, climate, existing natural vegetation, type of feed, administration of antibiotics and general management. In addition, another survey was designed to analyze productive parameters such as weight at slaughter, slaughterhouse input weight, channel weight, average daily gain and feed conversion rate. These surveys were completed in collaboration with the veterinarians of the association and the personnel of the slaughterhouse.

All statistical analyses were carried out using SAS^®^ V.9.1.3 (SAS institute Inc., Cary, NC, USA), setting *p* ≤ 0.05 as the significance level. For all analyses, the farm was considered the experimental unit. After conducting the surveys, the variables were coded, databases were elaborated in Excel^®^ and the relevant statistical comparisons were carried out. First, Shapiro–Wilk and Levene tests were applied to assess if the continuous variables followed a normal distribution. The variables with a non-normal distribution were compared by non-parametric tests (Wilcoxon test). In the case of categorical variables, a Chi-square test (χ^2^) or Fisher’s exact test was used to assess the possible association among the variables. The risk factors associated with *Salmonella* prevalence at the farm level were determined by considering the percentage of infected animals on the farm as a continuous variable (*n* = 180) or as a categorical variable, considering a 20% prevalence cut-off (i.e., low or high prevalence on farms showing *Salmonella* in less than or more than 20% of pigs, respectively). Moreover, the variables detected as relevant (*p* < 0.2) in the univariable analysis were assessed for associations, applying *X*^2^ or an exact Fisher’s test in the case of categorical variables and collinearity analysis in the case of continuous variables. Finally, only variables that showed no association were included in a multivariable analysis. Thus, *Salmonella* prevalence (continuous variable) on a farm was analyzed by multivariable regression analysis, whereas low or high prevalence (categoric variable) was analyzed by a logistic, multivariable analysis.

### 2.5. Microbiota Analyses of Cecum Content

A total of 35 IC samples were selected to analyze the intestinal microbiota composition in animals from farms with very high or null prevalence of *Salmonella* spp. Thus, 10 and 5 pigs positive for *Salmonella* from two different farms and 20 pigs from two negative farms (10 animals/farm) were selected for microbiota analyses. The selection of farms was based on geographical location, type of food and presence/absence of *Salmonella* on the premises.

### 2.6. Intestinal Content DNA Extraction, PCR Amplification and Microbiota Analyses

Bacterial DNA was extracted from 0.2 mg of each sample using the PowerSoil™ DNA isolation kit (MO BIO) under manufacturer’s conditions. DNA samples were treated with 100 μL of the elution buffer and stored at −20 °C until further processing. The V1–V2 regions of 16S rRNA genes were amplified with primer pairs F27 (5′-AGAGTTTGATCCTGGCTCAG-3′) and R338 (5′-TGCTGCCTCCCGTAGGAGT-3′). Both primers included sequencing adaptors at the 5′ end, and forward primers were tagged with different barcodes. The PCR mixture (50 μL) contained 2 μL DNA template (~5 ng), 5 μL of 10x AccuPrime™ PCR Buffer II, 0.2 μM of each primer and 1 U of AccuPrime™ Taq DNA Polymerase High Fidelity (Invitrogen, Life Technologies, Carlsbad, CA, USA). The PCR thermal profile was 2 min at 94 °C followed by 30 cycles of 1 min at 94 °C, 1 min at 55 °C, 1 min at 72 °C and a final extension of 7 min at 72 °C. To check the absence of reagents contamination, each PCR included a negative control. For each amplicon, both concentration and quality were determined using Agilent Bioanalyzer 2100. Samples were sequenced on an Ion Torrent Personal Genome Machine (PGM) with the Ion 318 Chip Kit v2 (Life Technologies) and the Ion PGM^TM^ Sequencing 400 Kit (Life Technologies) under manufacturer’s conditions. The raw sequences were deposited in NCBI under the Bioproject accession number PRJNA723169.

### 2.7. Quality Control, Operational Taxonomic Unit (OTUs), Diversity and Composition Analyses

Raw reads were imported into the EzBioCloud 16S-based MTP (microbiome taxonomic profiling) pipeline [19], which was used to analyze the whole dataset. Low quality sequences were filtered out using the following criteria: (i) read length less than 100 bp or more than 2000 bp; (ii) averaged *Q* value < 25; (iii) not predicted as a 16S rRNA gene by the hidden Markov model-based search; or (iv) detected as a singleton when sequences with at least 97% similarity did not match any of the reference sequences from the 16S database using the UCLUST program. Quimeric sequences were removed using the UCHIME program [20].

Taxonomic assignment was performed using the VSEARCH program [21] to detect and calculate the sequence similarities of the queried single-end reads against the EzBioCloud 16S database (version PKSSU4.0); 97% 16S similarity was used as the cut-off for species-level identification. Single-end reads from each sample were clustered into OTUs using the open-reference method [19].

Alpha diversity was estimated by the OTU richness of each sample, measured by abundance-based coverage estimation (ACE) and Chao1 methods. Diversity was estimated by the Shannon, Simpson and Phylogenetic Diversity indices. Statistical significance of the alpha diversity values between *Salmonella* -positive and -negative groups was assessed using the Wilcoxon rank sum test. A *p* value < 0.05 was considered statistically significant. A Venn diagram plot was drawn using Venn diagram software (available online: http://bioinformatics.psb.ugent.be/webtools/Venn/, accessed on 17 June 2021) to compare the species among *Salmonella* -positive and -negative groups.

Beta diversity was assessed with UniFrac distances based on taxonomic abundance profiles. Statistical significance of the beta diversity clustering by *Salmonella* -positive and -negative groups was assessed using permutational multivariate analysis of variance (PERMANOVA) (*p* < 0.05). UniFrac distance matrices were used to perform principal coordinate analysis (PCoA) with the Qiime2 tool [22] to compare the microbial communities among the groups. Additionally, differences in beta diversity were evaluated among farms.

Linear discriminant analysis (LDA) effect size (LEfSe) [23] was used to compare *Salmonella* -positive and -negative groups and to identify statistically significant differences in taxa abundances between groups. Taxonomic levels with LDA score > 2 and *p* value < 0.05 were statistically significant.

## 3. Results

### 3.1. Prevalence of Salmonella in the Extensive System

The presence of *Salmonella* in the IC from free-range pigs was studied using both the ISO as and PCR-invA methods and confirmed by the isolation and typing of the pathogen. The standard ISO allowed us to detect *Salmonella* in 47 out of 180 (26.1%) IC samples, whereas PCR-invA detected the pathogen in the same samples, as well as in an additional 11 (Appendix A). The concordance test between both diagnostic techniques was almost perfect (k = 0.851) due to the high number of negative samples in both techniques (Appendix A).

Considering both diagnostic methods, *Salmonella* was detected in the feces of 32.2% (58/180) of free-range pigs, allocated among 10 out of the 12 (83.3%) farms sampled (Table 1). Also, the mean prevalence within *Salmonella*-positive farms (i.e., herds containing at least one positive pig) was 38.7%, showing that for most of the farms (66.7%), more than 20% of the pigs shed *Salmonella* in their feces (Table 1 and Appendix A).

### 3.2. Serotyping and Antimicrobial Susceptibility

As shown in Table 2, the 58 *Salmonella* isolates belonged to 6 different serovars, being *S*. Typhimurium monophasic variant (4,[5],12:i:-) the most frequently found (46/58; 79.3% pigs) followed by far by *S*. Bovismorbificans (6/58; 10.3% pigs). Interestingly, the monophasic variant was found in 8/10 positive farms, being the only serovar circulating in 6 of these farms with the highest number of infected animals, varying between 46.7% and 86.7% (Table 2). Regarding antimicrobial susceptibility, 89.7% (52/58) of isolates exhibited resistance to at least one of the antimicrobials tested, being streptomycin (51/58; 87.9%), sulfisoxazole (44/58; 75.7%), ampicillin (31/58; 53.4%) and tetracycline (24/58; 41.4%), the drugs with the highest proportion of resistant isolates. Additionally, 58.6% (34/58) of them showed different multi-drug resistant profiles. Thirteen out of the 46 (28.2%) monophasic variants exhibited the ASSuT profile (Table 2), although other profiles were also detected such as the penta-resistant phenotype (ACSSuT).

### 3.3. Risk Factors Associated with the Presence of Salmonella in IC Samples

Once the questionnaires were filled out at the farms and slaughterhouse, variables were coded and data was debugged. Questions with no answers or answers without variability were removed from the statistical analysis. No significant differences were detected among the five productive parameters (mean ± SD) analyzed: piglet weight at the beginning of the fattening phase (29.3 kg ± 3.9), input weight at the slaughterhouse (154.3 kg ± 8.7), carcass weight (118 kg ± 6.6), daily gain mean (773.6 ± 54.3) and feed conversion rate (3.76 ± 0.238). As a result, a total of 52 variables were analyzed, and only those risk factors that showed *p* ≤ 0.2 in the univariable analysis were included in the multivariable analysis. Thus, four variables were associated with a high risk of detecting *Salmonella* in IC, considered categorical (Table 3); (i) large farms (>100 pigs); (ii) farms that used Feed A (with soybeans and beet pulp) versus Feeds B and C (beans and rapeseed meal, respectively); (iii) farms with exclusively grass-based vegetation versus those with chestnuts, acorns and beeches available; and (iv) farms with silos that were neither cleaned nor disinfected. These last three variables were associated with each other (*p* < 0.05), and the feed and extra vegetation available to the animals were selected to be included in the multivariable analysis. Details about these variables by farm are included as Appendix A (Appendix A). Thereafter, the multivariable analysis was performed by introducing into the model the “feed composition” (Feed A vs. B/C) and the “holding size” (>100 vs. ≤100 animals/farm) as independent variables. The logistic model indicated that feed A (soybeans and beet pulp) was a significant risk factor (*p* = 0.008) associated with the presence of *Salmonella* (Table 3).

### 3.4. Microbiota Analysis in Two Pig Populations

As shown in Appendix A, a total of 35 fecal samples were selected to study their microbiota composition, distributed between two groups of 15 *Salmonella*-positive pigs from two different farms (10 and 5 pigs/farm) and 20 *Salmonella*-negative pigs from two negative farms (10 animals/farm). A mean of 50,621 reads per sample were analyzed. A total of 21 phylum, 55 classes, 117 orders, 265 families, 987 genera and 3497 species were identified at a 97% sequence similarity level. A mean number of 2479 species were detected in the *Salmonella*-positive samples compared with 2910 detected in the *Salmonella*-negative samples. A total of 1897 OTUs were in both groups; however, 587 and 1018 were unique for *Salmonella*-positive and -negative samples, respectively (Figure 1A). No significant differences were observed in species richness between both groups using either the Shannon or Simpson tests (Table 4), but a significantly higher phylogenetic diversity index was obtained in the *Salmonella*-positive group (*p* = 0.044).

The analysis of microbial community composition showed that the predominant phyla in the *Salmonella*-positive and -negative groups were Firmicutes (47.79% and 49.01%, respectively) and Bacteroidetes (36.81% and 43.67%, respectively). The relative abundance of *Firmicutes* was similar in both groups, whereas the abundance of *Bacteroidetes* was significantly lower and the abundance of *Proteobacteria* was higher in the *Salmonella*-positive group (Figure 1B). The presence of 20 bacterial families was significantly differentiated among the two groups. The families more represented in the *Salmonella*-positive group were *Moraxellaceae*, *Planococcaceae*, *Ruminococcaceae*, *Enterobacteriaceae* and RF16_f in contrast to *Clostridiaceae*, AC160630_f and *Selenomonadaceae* in the *Salmonella*-negative group.

At the genera level (Figure 1C), *Prevotella* was the most abundant in both groups, with a relative abundance of 8.43% and 11.58% in the *Salmonella*-positive and -negative groups, respectively. *Sporobacter* (6.23%) was the second most abundant genus in the *Salmonella*-positive group and *Clostridium* (7.5%) in the *Salmonella*-negative. LEfSe analysis (LDA score > 2.0, *p* < 0.05) showed that taxa abundance was significantly different for 28 and 12 genera in *Salmonella*-positive and *Salmonella*-negative pigs, respectively (Figure 2A). *Acinetobacter*, EU843998_g, *Sporobacter*, *Caryophanon*, *Enterobacteriaceae*_uc, *Escherichia*, *Flavobacterium* and *Enterobacter* were represented in the microbiota of the *Salmonella*-positive group, whereas *Clostridium*, *Turicibacter*, AB494828_g and PAC001421_g, among others, were significantly represented in the *Salmonella*-negative group.

Finally, UniFrac-based PERMANOVA analysis indicated significant differences (*p* = 0.006) between *Salmonella*-positive and -negative groups. A three-dimensional PCoA graph (Figure 2B) based on UniFrac distances (ANOSIM value R = 0.05, *p* < 0.001) showed well-defined and different microbiota compositions for each group. Significant differences were also observed in beta diversity among farms classified as *Salmonella*-negative (*p* = 0.013); while *Clostridiales*, *Erysipelotrichales* and *Selenomonadales* were highly represented in farm 12, the *Spirochaetales* order was more abundant in farm 10.

## 4. Discussion

The prevalence of *Salmonella* in pigs reared in outdoor systems has been poorly studied. The actual re-emergence of this type of production makes it necessary to study the prevalence of infection and the risk factors associated with the occurrence of *Salmonella*. Furthermore, studies on the microbiota composition in these animals have not been previously reported; as well as, variations of microbiota composition with regard to the presence/absence of *Salmonella*. In general, high *Salmonella* prevalence was observed in animals reared outdoors but with high differences between farms. Several studies have been carried out in outdoor systems; however, they reported different management practices including organic, non-organic or Iberian pigs, making these results hardly comparable to ours [24,25]. Contrarily to our findings, only a single antibiotic-free pig from the extensive system was positive for *Salmonella* spp. [24], and 5.3% of Iberian pigs belonging to 33% of the herds from South Spain were *Salmonella*-positive in their mesenteric lymph nodes (MLN) [25]. These differences in prevalence might be explained by the type and quantity of samples (i.e., 1 g of MLN vs. 25 g of feces used in our study) and could also be influenced by different weather and feeding conditions. Jensen and collaborators investigated the dynamics of *Salmonella* infection in organic pigs and its dispersion in the pasture, demonstrating that *Salmonella* persisted in the environment, contaminating grass then able to cause infections in pigs [26]. The longitudinal occurrence of *Salmonella* in outdoor systems in the UK provided evidence that movement to a new location in outdoor systems has an overall beneficial effect on *Salmonella* carriage [27], decreasing its prevalence in feces from 29.6% to 16.9%. This management practice could help to reduce *Salmonella* in outdoor herds. However, new land is not always available, and perhaps more frequent resting and rotation of the land used for paddock systems within a field site could be of help. Finally, in the UK and Denmark [27], pigs reared in outdoor systems showed similar *Salmonella* prevalence to those of our study (29.6%). The prevalence of salmonellosis reported herein was very similar to that reported by Spain [28] and Aragón [29], both with intensive indoor systems. To our surprise, extensive swine showed a much higher prevalence than the intensive fattening pig of Navarra (8.4% of the animals) [12].

Knowledge of the serovars and AR of circulating *Salmonella* strains is necessary both, for appropriate antibiotic treatments and for epidemiological follow-up, e.g., identification of the origin of contamination and control of the expansion of clones carrying mobile genetic elements with multi-drug resistant profiles. The emergence of the monophasic variant exhibiting multi-drug resistance, ASSuT [30], has been increasingly reported in the last few years [31] and is widely disseminated in the EU [32,33], including Spain (this study). The spread of multi-drug resistant *Salmonella* from farm to fork implies a major risk from a one-health perspective. For this reason, detecting risk factors associated with the presence of *Salmonella* in the primary sector is essential for designing mitigation strategies. Additionally, it is interesting that only the monophasic variant was recovered herein, while *Salmonella* Typhimurium (4,5:12:i:1,2) was the most prevalent serovar detected in fattening pigs and sows within intensive systems [11,12,29]. Furthermore, the most common serovars reported in Iberian pigs were Anatum and Typhimurium with resistance to streptomycin and tetracycline [25], differing from those described in our study.

The control of *Salmonella* requires determining those risk factors associated with a high prevalence in fattening pigs reared in outdoor systems. The number of farms analyzed herein (12 out of 32) were enough to determine the prevalence of *Salmonella*; however, the multivariable analysis was performed only with those variables that showed highly significant *p* values in the univariable model. Unfortunately, there were many variables included in the analysis, but not enough farms (low statistical potency). Moreover, several risk factors showed statistical association between them, impeding the evaluation of the relative weight of each variable in the model. Further work should be performed on those farms that have sufficient variability in associated parameters (feed compositions, different extra feed, silo cleaning) to validate the model.

Diet A (soybeans and beet pulp) was a risk factor that increased the prevalence of *Salmonella*. Perhaps there was defective or insufficient heat treatment of the granulate or cross-contamination during processing, storage and/or transportation that favored the growth of the pathogen. In addition, feed contamination may come from silos where inadequate cleaning and disinfection is carried out, especially if the silos are exposed to rodents, birds, insects and other free-living vectors [27,34,35]. On the other hand, the origin of the feed could also determine the quality of the ingredients (raw materials and/or additives) used in its formulation and physical structure. This aspect, associated to the nutritional quality of the surrounding vegetation, probably modulates the composition of the bacterial populations in the digestive tracts of pigs from extensive systems. Pigs fed with more nutritious products could develop a microbiota capable of preventing the presence of *Salmonella* in the intestine by creating a suitable microenvironment (digestive tract acidification) and/or the proliferation of other bacterial taxa that could promote competitive exclusion of the pathogen. To address this hypothesis, we analyzed the profile of the intestinal microbiota of a selected population with different types of food and the presence and absence of *Salmonella*.

The gut microbiota in pig cecum samples was largely composed of obligate anaerobic bacteria belonging to *Firmicutes* and *Bacteroidetes*, the main phyla detected herein, accounting for more than 88% of all bacteria in pigs, as observed by other authors [8,9]. These phyla degrade the nutrients present in the distal gut to a variety of metabolites producing short-chain fatty acids (SCFAs), with acetate, propionate and butyrate being the most abundant [36]. Interestingly, several studies have demonstrated the effects of propionate and butyrate production in the prevention of *Salmonella* colonization [37,38,39]. In fact, our results confirm higher abundance of *Clostridium*, *Turicibacter*, Bacteroidaceae_uc, and *Lactobacillus* in *Salmonella*-negative animals, all genera associated with the production of these two SCFAs [40]. Additionally, species of *Lactobacillus* have been proven to effectively inhibit the colonization of *Salmonella* by the secretion of metabolites with antimicrobial activity or by impairing the adhesion of *Salmonella* to the intestinal cells [41]. Contrarily, acetate production has been linked to *Salmonella* invasion [39] by inducing the expression of the genes located in pathogenicity island I (ISPI-1) required to penetrate the intestinal epithelial cells [42]. In this regard, we detected an enrichment of the genus *Sporobacter*, described as an acetate producer [43] in the *Salmonella*-positive group, together with *Acinetobacter*, which can utilize acetate as carbon source. In fact, an increase in the abundance of *Proteobacteria*, a phylum that has been associated with a microbial signature of dysbiosis in the gut and epithelial injury, was observed in *Salmonella*-positive animals [44,45]. Families of *Moraxellaceae* and *Enterobacteriaceae* belonging to the Gammaproteobacteria class were enriched in these animals, with the latter family containing a large number of potential pathobionts for humans and pigs including *Salmonella*, *Escherichia* and *Enterobacter*. Herein, V1–V2 amplification of the 16S rRNA region allowed us to detect differences in the abundance of *Proteobacteria* among *Salmonella* prevalence groups, despite the suggested limitation proposed by Johnson and collaborators [46]. Differences in beta diversity were also observed among pigs from farms negative for *Salmonella*, probably due to the dietary differences associated with a diet rich in carbohydrates and fiber in animals feeding on acorns as well as starch, provided mainly by the chestnuts.

The results obtained in this study indicated that diet conditioned the intestinal microbiota of pigs, a decisive factor in the presence/absence of *Salmonella* in the intestinal contents of these animals. Along these lines, future studies could contribute to elucidating the impact of different dietary ingredients, comparing those traditionally of good quality (barley, wheat and maize) with others containing large numbers of raw materials (soybeans, sorghum, cassava, pulp or beet molasses), which could potentially balance the porcine intestinal microbiota and thereby prevent the presence of *Salmonella* in the intestinal tract.

## Figures and Tables

**Figure 1 foods-10-01410-f001:**
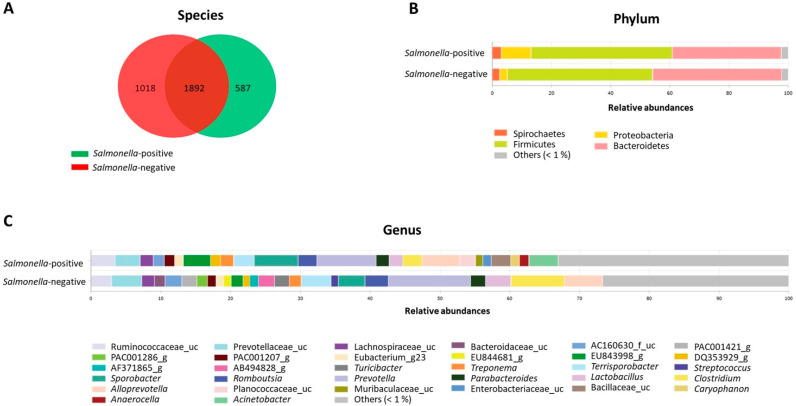
Microbial community comparisons between *Salmonella*-positive and -negative groups. (**A**) Venn diagram showing unique and shared species. (**B**) Relative abundance of bacterial phyla. (**C**) Relative abundance of bacterial genera.

**Figure 2 foods-10-01410-f002:**
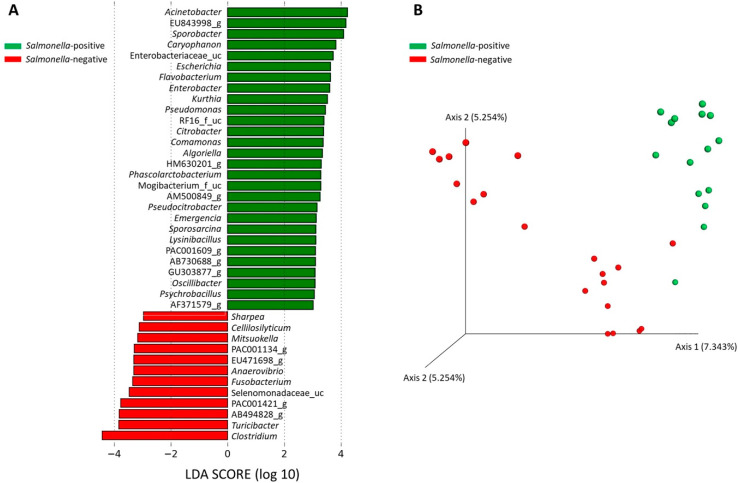
Differential bacterial taxa analysis in *Salmonella*-positive (green) and *Salmonella*-negative (red) groups. (**A**) Distinctive genera of each group obtained by LEfSe analysis (LDA score > 2.0; alpha value < 0.05). (**B**) Principal coordinate analysis (PCoA) graph based on UniFrac distances.

**Table 1 foods-10-01410-t001:** Detection of *Salmonella* by the ISO 6579:2002/Amd 2007 and/or molecular methods in the intestinal content of free-range pigs of North Spain.

*Salmonella* spp. Isolation	Intestinal Content
No. (%; CI95) ^1^ positive pigs ^2^/total pigs	58/180 (32.2%; 25.8–39.3)
No. (%; CI95) ^1^ positive farms ^2^/total farms	10/12 (83.3%; 55.2–95.3)
No. (%; CI95) ^1^ positive pigs ^2^/pigs in positive farms	58/150 (38.7; 31.2–46.6)
No. (%) farms above 20% prevalence/total farms	8/12 (66.7%; 39.1–86.2)

^1^ mean % and 95% of confidence interval; ^2^ positive pigs or farms were those where at least 1 *Salmonella* positive sample was detected.

**Table 2 foods-10-01410-t002:** Prevalence of *Salmonella* in the intestinal content of free-range pigs. Data are presented by farms, serovars and antimicrobial resistance (AR) profiles of the isolates found.

Farm Code	No. (%) Samples Positive/Total	Serovars (No. of Strains)	AR Profiles ^1^ (No. of Strains)
1	13/15 (86.7%)	4,5,12:i:- (13)	SSu (5); ASSuT (4); ASSu (2); ACSSuT (2)
2	8/15 (53.3)	4,5,12:i:- (8)	ACS (3); CS (2); ACSSu (1); AS (1); ACSSuT (1)
3	7/15 (46.7%)	4,5,12:i:- (7)	SSu (4); ASSuT (3)
4	7/15 (46.7%)	4,5,12:i:- (7)	ASSu (4); ASSuT (2); SSu (1)
5	6/15 (40%)	4,5,12:i:- (5); *diarizonae* (1)	ASSuT (3); SSu (1); A (1) Susceptible (1)
6	5/15 (33.3%)	Bovismorbificans (4); Altona (1)	ACSSuT (2); CSSuT (2); S (1)
7	4/15 (26.7%)	Bovismorbificans (2); Meleagridis (2);	CSSuT (2); Susceptible (2)
8	4/15 (26.7%)	4,5,12:i:- (2); Amsterdam (1); Altona (1)	ASSuT (2); Susceptible (2)
9	3/15 (20%)	4,5,12:i:- (3)	ASSu (2); Susceptible (1)
10	1/15 (6.7%)	4,5,12:i:- (1)	ASSuT (1)
11	0/15 (0%)	N.A.	N.A.
12	0/15 (0%)	N.A.	N.A.
Total	58/180 (32.2%)	6 serovars (58)	11 AR profiles

^1^ A: ampicillin; C: chloramphenicol; S: streptomycin; Su: sulfisoxazole and/or trimetoprim–sulfometoxazole; T: tetracycline; N.A.: not applicable.

**Table 3 foods-10-01410-t003:** Risk factors associated with salmonellosis in free-range pigs. The prevalence of *Salmonella* on the farm was considered to be a continuous variable or as a categorical variable considering the cut-off of 20% prevalence (low or high prevalence on farms with presence of *Salmonella* spp. in less than or more than 20% of pigs, respectively). The analysis of the prevalence as a continuous variable or as a categorical was carried out by a multivariable regression or logistic analysis, respectively.

Risk Factors	Statistical Analysis (*p* Values)			
	Univariable		Multivariable	
	Prevalence	Low/High	Prevalence	Low/High
(i) Large farms (>100 animals)	0.02	0.06	0.11	0.19
(ii) Feed with A brand	0.13	0.02	0.29	0.008
(iii) Extra feed with grass	0.38	0.02	N.A.	N.A.
(iv) Neither cleaning nor disinfection of silos	0.04	0.02	N.A.	N.A.

N.A.: not applicable because these variables were associated to risk factor “ii”.

**Table 4 foods-10-01410-t004:** Operational taxonomic units (OTUs)’ richness and alpha diversity indexes estimated in the gut microbiota of *Salmonella*-positive and *Salmonella*-negative pigs.

Parameter	OTU Richness and Alpha Diversity Indexes Median and (IQR) ^a^ Values		
	*Salmonella*-Positive Pigs	*Salmonella*-Negative Pigs	*p* Values
OTU number	3132 (583)	3047 (714.73)	0.271
Chao1	3156.06 (581.84)	3085.74 (725.19)	0.301
ACE ^b^	3227.75 (580.90)	3166.53 (725.19)	0.301
Shannon	6.44 (0.64)	6.04 (0.80)	0.072
Simpson	0.01 (0)	0.01 (0.01)	0.062
Phylogenetic diversity	1366 (80)	1288 (119.50)	0.044 ^c^

^a^ IQR: interquartile range; ^b^ ACE: abundance-based coverage estimation; ^c^
*p* < 0.05 between both pig groups by Wilcoxon rank sum test.

## Data Availability

All data generated in this study is included in the article. The raw sequences were deposited in NCBI under the Bioproject accession number PRJNA723169. Further information on data and samples is available from the corresponding author on request.

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
