# Peer review of "Prevalence of Salmonella in Free-Range Pigs: Risk Factors and Intestinal Microbiota Composition"

_foods, 2021, doi:10.3390/foods10061410_

Round 1

Reviewer 1 Report

The title is misleading: the work focuses on Salmonella in free-range pigs and limited considerations are made on the role of the microbiota.

The introduction focuses on Salmonella and pig farming management without any consideration of the role of the microbiota in relation to the pathogen although in line 72 " tha main goal of this study is to widen the knowledge available on the microbiota of pigs raised in extensive systems "
I believe that the introduction should be revised in relation to the  objectives identified.

Microbiota analyses was performed using V1-V2 regions of 16S rRNA gene: I ask you to justify the choice of this region;   V1–V2 region performed poorly at classifying sequences belonging to the phylum Proteobacteria (see Johnson et al, 2019), V1-V2 region may not be fully suitable for characterizing the gut microbiota.

The results show a high prevalence of Salmonella in extensive pig farming especially in farms with more than 100 animals and in relation to the composition of feed. However, there are no elements to support the hypotheses (ie feed analyses or the evaluation of the space available per animal or or the level of application of biosecurity standards in Salmonella positive and negative farms). Similarly, there is no discussion of the spread of AMR in Salmonella isolates.

The considerations on the role of the type of feed on the composition of the microbiota in relation to Salmonella are interesting, but do not clarify the context with respect to the focus of the work which is the free -range pigs

Author Response

Reviewer 1:

  1. The title is misleading: the work focuses on Salmonella in free-range pigs and limited considerations are made on the role of the microbiota.

We have changed the title, reading now as follows:

Prevalence of Salmonella in free-range pigs: risk factors and intestinal microbiota composition

  1. The introduction focuses on Salmonella and pig farming management without any consideration of the role of the microbiota in relation to the pathogen although in line 72 "the main goal of this study is to widen the knowledge available on the microbiota of pigs raised in extensive systems". I believe that the introduction should be revised in relation to the objectives identified.

Thank you for the comment, we agree with the reviewer. Accordingly, we have included in the Introduction section the following:

Recent studies have shown that Salmonella enterica is a pathogen capable of causing alterations to the composition of the intestinal microbiome, reducing the diversity and abundance of species regarded as beneficial, such as lactic acid bacteria [8,9]. However, most of these studies are based in experiments where pigs have been challenged with the pathogen, and very few studies have focused on natural exposure of pigs to the pathogen; this is especially true for free range pigs.

  1. Microbiota analyses was performed using V1-V2 regions of 16S rRNA gene: I ask you to justify the choice of this region; V1–V2 region performed poorly at classifying sequences belonging to the phylum Proteobacteria (see Johnson et al, 2019), V1-V2 region may not be fully suitable for characterizing the gut microbiota.

We agree with the reviewer that other combinations (eg. V3-V4) could perform better than V1-V2 for Proteobacteria, which could underestimate the abundance in these taxa. However, our sequencing service had the library optimized for V1-V2 and we accepted this method because, at least in our case, it allowed us to discriminate two well-separated populations. This interesting point and the suggested reference (Johnson et al., 2019) has been introduced in the Discussion and References sections (Lines 447-449) as follows:

Herein, V1-V2 amplification of the 16S rRNA region allowed to detect differences in the abundance of Proteobacteria between groups, despite the suggested limitation proposed by Johnson et al, 2019 [46].

  1. The results show a high prevalence of Salmonella in extensive pig farming especially in farms with more than 100 animals and in relation to the composition of feed. However, there are no elements to support the hypotheses (ie feed analyses or the evaluation of the space available per animal or the level of application of biosecurity standards in Salmonella positive and negative farms). 

As stated in the Material & Methods section, “Risk factors associated with Salmonella presence in pigs’ IC were studied by performing an epidemiological survey with 73 variables divided into five sections. The variables analysed were related to the geographical location of farms, climate, existing natural vegetation, type of feed, administration of antibiotics and general management.” This statement includes other variables such as (but not only) the space available per animal as well as the level of application of biosecurity standards. Thus, the variables suggested by the reviewer and other variables were not significantly associated in a univariable analysis to be also included in the multivariable one.

  1. Similarly, there is no discussion of the spread of AMR in Salmonella isolates.

Spread of AMR in Salmonella isolates could imply a risk from a one-health perspective. For this reason, this kind of studies could be very valuable to decide if any mitigation measures could be necessary to tackle this risk. A comment has been included in Lines 391-393, as follows:

Spread of multi-drug resistant Salmonella from farm to fork implies a major risk from a one-health perspective. For this reason, detecting risk factors associated to the presence of Salmonella in the primary sector is essential to design mitigation strategies. Additionally,…”

  1. 6. The considerations on the role of the type of feed on the composition of the microbiota in relation to Salmonella are interesting, but do not clarify the context with respect to the focus of the work which is the free-range pigs.

As stated in Results section, we found an association between the diet (i.e. type of feed and natural supplement available) and salmonellosis prevalence of free-range pigs. In fact, as it can be seen in Table S2, feeding based on dry feed type A (with soybeans and beet pulp) and extra diet of grass was associated to higher prevalence. However, the type of feed cannot be directly associated the composition of the microbiota of free-range pigs, since this extensive farming system (in contrast to intensive farms) allows animals to supplement their diet with different raw material obtained ad libitum from the surrounding ecosystem. It would have been ideal to have free-range pigs with a range of different diets but this was not the case. Thus, free-range pigs and diet are associated and it cannot be disentangled.

Sincerely,

Dr. María-Jesús Grilló

Reviewer 2 Report

The study assessed the prevalence of Salmonella in the intestinal content of pigs reared in farms with extensive systems in Spain. A total of 180 specimens from 12 farms were examined to find 32.2% of animals and 83.3% of farms were Salmonella positive. Serovar monophasic Typhimurium with antimicrobial resistance was predominant. Risk factors for Salmonella prevalence were also assessed to find feeding and cleaning were of significance. The intestinal microbiota were analyzed for Salmonella-positive/negative specimens. Butyrate and propionate producers and acetate producers were rich in Salmonella-negatives and -positives, respectively.

The manuscript is well written. The study is clear.

Regarding the assessment of risk factors in Table 3, how was the p values if Salmonella-negative/positive would be applied instead of the criteria Low/High. If they are comparable, criteria negative/positive would be more easily understood than Low/High.

Minor

  1. Line 124-, abbreviations of antimicrobials are not clear. For example, “A” stands for ampicillin “and/or” amoxicillin-clavulanic acid?

Author Response

Reviewer 2:

  1. Regarding the assessment of risk factors in Table 3, how was the p values if Salmonella-negative/positive would be applied instead of the criteria Low/High. If they are comparable, criteria negative/positive would be more easily understood than Low/High.

Classification of farms as “low/high Salmonella prevalence” should be maintained since, otherwise, only two farms would be negative for Salmonella, and the risk assessment analyses could not be performed.

  1. Line 124-, abbreviations of antimicrobials are not clear. For example, “A” stands for ampicillin “and/or” amoxicillin-clavulanic acid?

As in previous epidemiological works, we have used A to abbreviate aminopenicillins. We tested both ampicillin and amoxicillin-clavulanic acid, but really all strains resistant to ampicillin were considered as “A”. Accordingly, amoxicillin-clavulanic has been removed from the manuscript, in Mat&Met section (Line 132) and in Table 2 (footnote).

Sincerely, 

Dr. Maria-Jesús Grilló